# A Study on the Influence of Recreational Activities Intervening in Natural Science Courses on Learning Motivation and Learning Outcomes—The Case of Tabletop Games

Chao-Yen Lin [1], Hsiao-Hsien Lin [2,*] , Kuo-Chiang Ting [3], Chih-Chien Shen [4] , Chih-Cheng Lo [5,*] , Hsiu-Chu Hung [5] and Li-Ju Tsai [5]

1 School of Physical Education, Jiaying University, Meizhou 514015, China
2 Department of Leisure Industry Management, National Chin-Yi University of Technology, Taichung City 411030, Taiwan
3 Department and Graduate Institute of Physical, Education, University of Taipei, Taipei City 10048, Taiwan
4 Institute of Physical Education and Health, Yulin Normal University, 1303 Jiaoyu East Rd., Yulin 537000, China
5 Department of Industrial Education and Technology, National Changhua University of Education, Changhua City 500207, Taiwan
* Correspondence: chrishome12001@yahoo.com.tw (H.-H.L.); charleslo@cc.ncue.edu.tw (C.-C.L.)

**Abstract:** This study explored the effects of the integration of tabletop games in natural science and environmental education courses on students' learning motivation and learning effectiveness. Four tabletop games were designed and integrated in educational programs, and 100 fifth-grade students at an elementary school in Taiwan participated in the study. Participants were divided into experimental and control groups. The former received an instructional program with tabletop games; the latter was the original program. Both groups had received the instructions for the two units of the natural science course for eight weeks. Furthermore, participants' academic performance results were collected to investigate the effects of tabletop games on their learning effectiveness. This study has shown that integration of tabletop game in the natural science course significantly improved students' learning attention. While the academic performance of the experimental group improved, the standard deviation decreased. In addition, qualitative data indicated the high acceptability of tabletop games integration in courses. Tabletop games integration in courses might also enhance different learning motivational dimensions. The findings of the integration of tabletop games might be beneficial to educational curriculum-specific pedagogy.

**Keywords:** educational sustainability; tabletop game; learning motivation; learning effectiveness; environmental education; flipped classroom

## 1. Introduction

With the rapid development of information technology, children's addiction to the internet in an online virtual environment not only affects their physical and mental health, but also impacts their regular schedule [1]. Given such rapid changes in technology, ideas, and concepts, we should aim to attract children's minds by designing curriculum and content that maintains their attention. Teachers must apply flexible and systematic thinking in line with the current situation and take students as a center that needs to be motivated to promote their long-term memory and computational thinking [2,3]. However, teachers are required to prepare for lessons beforehand, and, therefore, use their wisdom to flexibly grasp of students' attention and integrate suitable tabletop games into the curriculum before teaching to enhance students' learning motivations and effectiveness. This paper investigates the effects of integrating tabletop games into a natural science course on the learning motivation and effectiveness of fifth-grade elementary school students.

Tabletop games are defined mainly as table-based games with rules designed by the creators based on the aim of course content. Tabletop games are classified into eight

categories: abstract games, fetch games, children's games, family games, party games, strategy games, themed games, and war games [4]. Tabletop games can be used as fun way to provide a natural context for interpersonal interactions and communication. Therefore, this study uses tabletop games suitable for children and creates group games related to the curriculum, integrating comprehension and responsiveness.

Tabletop games have numerous applications for students' learning, and many studies have been conducted on their application in teaching or as teaching materials in the classroom [1,5–7]. However, few empirical studies have investigated their integration into different units of courses in the field of natural science. The main motive of this study is to integrate tabletop games into teaching activities to enhance the teaching of natural science courses. We designed four tabletop games for integration into the curriculum: "Heart Attack" and "Back to Earth" are integrated into The Sun unit; "Big Mouth" and "Operation to Rescue Taiwan's Native Species" are integrated into The World of Plants unit.

This paper intends to use tabletop games as a teaching method to enhance students' continuous learning motivation. We employed the games in the two units of natural science courses: The Sun and The World of Plants. Teaching strategies incorporating these games are expected to enhance students' learning motivation and effectiveness. This study employs an experimental design to explore the difference in the learning effectiveness and motivation of experimental and control groups of students when tabletop games are integrated into natural science courses.

### 1.1. Learning Motivation and Teaching Strategies

Motivation is considered the psychological driving force behind human behavior [8,9] and the internal process of guiding an individual toward a goal [10]. Motivation not only drives an individual to engage in a particular activity, but also provides an incentive for an individual to face the challenge [10,11]. Since learning motivation covers a wide range of theory, such as the attribution theory and the self-efficacy theory [12], learning motivation is defined as the intrinsic motivation that causes and maintains students to engage in coursework learning activities voluntarily [13,14].

The ARCS Motivational Teaching Model was developed by [15] as an integrated motivation theory and instructional design to integrate the expectancy-value theory, the attribution theory, the achievement theory, etc. The learning motivation theory should also include effectiveness, efficiency, and engagement [16]. The model focuses on the assimilation of an individual's internal factors (e.g., personal values, abilities, and cognitive values) and factors external to the learning environment (e.g., instructional design). The ARCS model was well-developed and the model is widely used [15,16].

The four components of the ARCS Model include Attention, Relevance, Confidence, and Satisfaction [15]. The motivation model should first arouse students' attention and interest in the course, then enable the students to discover that the course is personally relevant to them, so that students have confidence in themselves. Then students are capable of completing class tasks and attaining learners' satisfaction [16]. However, the actual teaching environment is extremely dynamic, and the individual differences of students are more important than the single teaching mode. Therefore, when using the ARCS motivation model, educators should assess students' needs in the light of the actual situation, particularly in the era of E-learning design [17]. The design of course curriculum should be designed according to students' needs so as to motivate them and maintain their learning progress. There is a wealth of literature on the application of the ARCS motivation model to teaching, and it has been applied to a variety of domains and stages [15,18]. In this study, we mainly use the ARCS motivation model as a research model to explore the motivation patterns of school children.

### 1.2. The Effectiveness of Integrating Tabletop Games in Natural Science

Teachers often need to understand and analyze students' learning status to adjust the curriculum and teaching activities [1]. Learning effectiveness refers to the cogni-

tive, affective, and skill-oriented improvement or change in learners after teaching and learning [19], and it also refers to the degree to which learners have successfully mastered the learning content. Learning effectiveness demonstrates learners' abilities, usually the ability to acquire knowledge, which also includes other concepts, such as curriculum or teaching materials, acquired knowledge, experience or skills, and learning outcomes [20]. Additionally, learning effectiveness refers to the actual changes produced by students both before and after receiving education, including the assessment results of school assignments and the progress in learners' knowledge, skills, and attitudes after teaching [21,22]. However, Hortigüela Alcalá et al. [23] stated that learning effectiveness is the effectiveness of tests administered on learners after teaching in order to understand what students have learned and to co-assess this. To summarize, learning effectiveness is an evaluation for teachers to understand the effectiveness of their teaching and the changes in students' knowledge, experience, or skills after a learning activity.

Teaching and learning are two sides of the same coin, and the core of learning effectiveness assessment lies in ascertaining the learner's knowledge level as well as their participation in the learning process [23]. A positive and active learning attitude and good learning motivation is of considerable benefit to students' learning and effectively enhances their learning effectiveness [24]. In this study, we will investigate whether the incorporation of tabletop games into the curriculum has a positive effect on students' learning effectiveness.

In summary, literature related to the effectiveness of learning in the area of natural science in elementary schools is scanty [25], and few studies have integrated tabletop games into a curriculum related to the Sun or plants in an elementary school. Accordingly, this paper intends to analyze the effects of tabletop games on the different stages of learning motivation of ARCS with different modules, which will teach us more about the application of tabletop games in teaching.

### 1.3. Aims and Research Questions

Board games generally are used for recreational activities. In education, games enable students to have a better understanding of the relationship of the content, process, and context of a subject matter [26]. Games in education are highly related to participants' learning motivation and teachers' teaching strategies. Accordingly, the aim of this paper is to investigate learning motivation with the effectiveness of integrating tabletop games. The research questions are as follows. How do tabletop games integrate into the teaching of natural science, specifically the Sun and plants, in elementary schools? How do the integration of tabletop games into the teaching of natural science influence students' learning motivation and learning effectiveness?

## 2. Research Method

### 2.1. Research Design

This study adopted a quasi-experimental research method, supplemented by qualitative data, such as interviews, feedback forms, and classroom observations. In terms of learning motivation, the pre-test was administered one week before the integration of tabletop games into teaching activities, and the learning motivation scale (LMS) was also adapted by the researcher. As for the learning effectiveness, the results of formative and summative assessments were used to statistically test and compare the impact of the integration of tabletop games on students' learning effectiveness. Additionally, after the integration of tabletop games into the teaching of The Sun and The World of Plants units of science and technology, students were asked to complete a learning feedback form, which was supplemented by qualitative data, such as interviews and classroom observations, in order to understand the effects of the integration of table games into teaching on the learning outcomes of students.

This study used expert validity at the beginning of empirical studies. First, we asked the supervising professor to review the guidance, and then invited 2 experts to discuss the content, structure and rules of the preliminary draft of the table game and the timing of

integration. Both experts have 10 years of teaching and research experience in universities. In addition, a previous study showed the reliability of ARCS questionnaires' Cronbach alpha are all above 0.834 [18]. Secondly, after the first draft of qualitative and quantitative measurement was completed, we further confirmed the questions of the scale according to the opinions provided by the experts. In addition, all interviews were conducted in Mandarin and translated by researchers while writing the report.

The experimental group followed [15] theoretical foundation of the ARCS model to integrate tabletop games at different points during the experiments. The experimental group appropriately integrated tabletop games in Activities 2 and 3 of The Sun and Activities 1 and 3 of The World of Plants units (Table 1).

**Table 1.** Tabletop game teaching, planning, and research design.

| Teaching Unit | Tabletop Game Name | Timing of Integration into the Course |
|---|---|---|
| ARCS learning motivation pre-test | | |
| Unit I: The Sun<br>Activity 1: The position of the Sun during the day | | |
| Unit I: The Sun<br>Activity 2: Changes in sunrise and sunset in the four seasons | Heart Attack | Attention |
| Unit I: The Sun<br>Activity 3: The impact of the three suns on life | Back to Earth | Confidence |
| Formative assessment-1 | | |
| Unit II: The World of Plants<br>Activity1: Plant structure and function | Big Mouth | Relevance |
| Unit II: The World of Plants<br>Activity 2: Plant reproduction | | |
| Unit II: The World of Plants<br>Activity 3: Classification of plants | Operation to Rescue Taiwan's Native Species | Satisfaction |
| Formative assessment-2 | | |
| Summative assessment | | |
| ARCS learning motivation post-test | | |

### 2.2. Research Framework

The independent variable of this research was the integration of tabletop games into the curriculum design, and the curriculum design was divided into two groups: experimental group and control group. In addition, the dependent variable was learning motivation and learning effectiveness. The architecture is shown in Figure 1.

1.  Independent variable: The independent variables of this study were four tabletop games: Heart Attack, Back to Earth, Big Mouth, and Operation to Rescue Taiwan's Native Species. These games were developed by the researchers and integrated into the curriculum for teaching.
2.  Dependent variable: The dependent variables include the learning motivation of students in the field of natural science, and the researchers' self-compiled evaluation of learning effectiveness. The evaluation of learning effectiveness was both formative and summative. Based on the performance of school children on the LMS, this paper compares and analyzes the changes in the pre-test and post-test scores of school children, and summarizes and analyzes the influence of tabletop games in the learning of nature and life technology on this research. Regarding the Assessment of Learning Effectiveness, comparisons and analyses are based on the performance of academic performance in formative and summative evaluations.
3.  Control variables: In order to control the factors that may affect the results of the experiment, except the independent variables, the control variables in this study included the teacher, teaching time, and teaching materials.

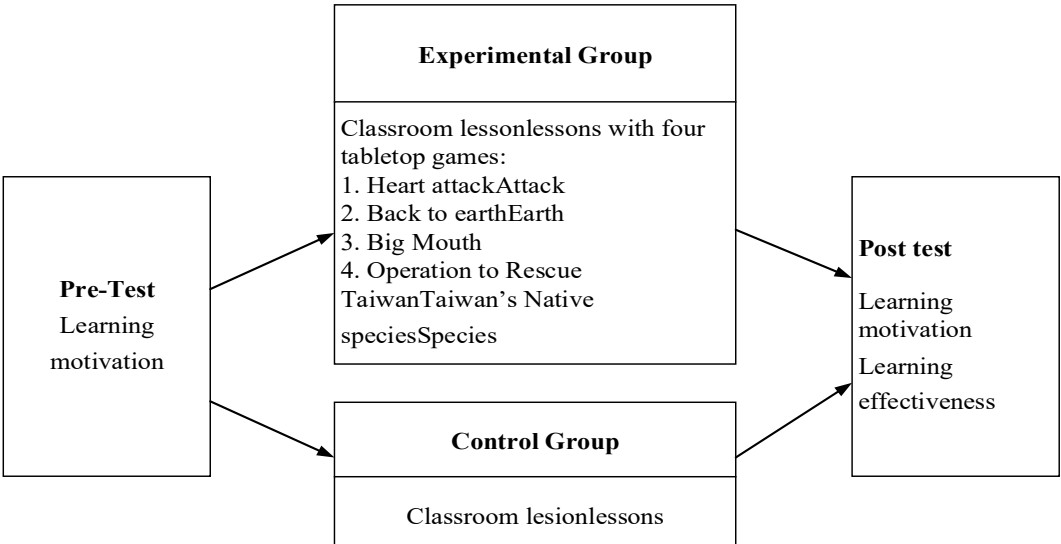

**Figure 1.** Research framework.

*2.3. Research Participants*

The participants of this study included students of the same age in four ordinary classes of grade 5 in a county elementary school in Taiwan. The children were divided into an experimental group and a control group, each containing two classes. There were 48 participants in the experimental group and 52 in the control group—100 in total. The experimental group accepted the teaching of tabletop games integration courses, whereas the control group did not play any of the four tabletop games in the classroom. The students in these four classes were selected based on their total scores in the fourth grade and the result of the S-type normal class placement of the academic affairs system. After the experimental stage, the study investigated whether the students in the two groups had different levels of changes in their learning motivation and effectiveness due to the integration of tabletop games in the curriculum.

The results of the statistical breakdown of the basic data are shown in Table 2 below, which clarifies that there is a difference of 10 persons between males and females in the official sample, and the number of male students exceeds female students. All the participants had experience of playing tabletop games and their knowledge and acceptance of the games was expected to be high percentage, which shows that tabletop games have become popular in the elementary school students.

**Table 2.** Summary of the sample of students used in this study.

| Background Variable | Category | Number of People | Percentage (%) |
|---|---|---|---|
| Gender | Male | 55 | 55 |
| | Female | 45 | 45 |
| Family members | 1–3 people | 9 | 9 |
| | 4–6 people | 58 | 58 |
| | 7 people or more | 33 | 33 |
| Father's degree | Below middle school | 7 | 7 |
| | High school | 47 | 47 |
| | Above university | 44 | 44 |
| Mother's degree | Below middle school | 8 | 8 |
| | High school | 48 | 48 |
| | Above university | 42 | 42 |
| Have you played a tabletop game? | Yes | 100 | 100 |
| | No | 0 | 0 |

## 2.4. Tabletop Game Teaching Design and Research Process

In this study, the tabletop games were integrated into the course four times, for 10–15 min each time. Before playing, rules were introduced for students to acquire a preliminary understanding of the playing method. This paper designed games that arouse learning motivation and designed the rules to be simple and intelligible. After each game, students were invited for an interview to share their learning experiences and influences. Table 3 shows the syllabus of teaching design for tabletop games.

**Table 3.** Teaching Design Syllabus for Tabletop Games.

| Times | Course Name | Learning Motivation | Teaching Design |
|---|---|---|---|
| 1 | Heart Attack assimilates into Unit I: The Sun Activity 2 | Attract attention | 1. Know the positions and immediate changes of the Sun during a day<br>2. Have used a compass and a solar observation box to measure the azimuth and altitude of the Sun<br>3. The purpose of explaining the rules of Heart Attack is to attract the attention of students so the gameplay remains simple and the content is related to The Sun unit<br>4. Students pose questions and teachers respond<br>5. Actual tabletop game operation<br>6. Identifying and solving problems<br>7. Interview |
| 2 | Back to Earth assimilates into Unit I: The Sun Activity 3 | Build confidence | 1. Know the positions and shadow changes of the Sun during a day, and use the north arrow and a solar observation box to measure the Sun's azimuth and altitude<br>2. Understand the changes of the Sun during the four seasons and the influence of the Sun on life<br>3. The purpose of explaining the rules of Back to Earth is to build students' confidence, so the gameplay remains simple and intelligible and related to the students' life experiences. The designed questions are consistent with the course content. After answering a question, students can proceed to the next level. Those who return to the Earth the fastest will win<br>4. Students pose questions and teachers respond<br>5. Actual tabletop game operation<br>6. Identifying and solving problems<br>7. Interview |
| 3 | Big Mouth assimilates into Unit II: The World of Plants Activity 1 | Personal relevance | 1. Know and understand the parts and functions of plants<br>2. Explain the rules of Big Mouth. The content on the game cards is related to students' lives to facilitate contextual awareness<br>3. Students pose questions and teachers respond<br>4. Actual tabletop game operation<br>5. Identifying and solving problems<br>6. Interview |
| 4 | Operation to Rescue Taiwan's Native Species assimilates into Unit II: The World of Plants Activity 13 | Create satisfaction | 1. Know and understand the parts and functions of plants<br>2. Know how plants reproduce and learn to classify plants<br>3. Explain the rules of Operation to Rescue Taiwan's Native Species. The purpose is to make students feel fulfilled and satisfied, so the content is related to The World of Plants unit. The fastest three cards will win.<br>4. Students pose questions and teachers respond<br>5. Actual tabletop game operation<br>6. Identifying and solving problems<br>7. Interview |

To sum, up, the tabletop games in this study were designed to arouse students' interest, hold their attention, build their confidence, and help them gain satisfaction. The mechanisms and conduct of the games were used to combine the learning contents of Unit I: The Sun and Unit II: The World of Plants for teaching activities.

## 2.5. Instrument

(1)　Learning the Motivation Scale

This paper adopted the Instructional Materials Motivation Survey Instrument (IMSI) of Dr. John Keller [15] of the Florida State University, and the LMS developed by the ARCS model as the theoretical bases for the study. The study used the Chinese version of the questionnaire and further revised to make the questionnaire intelligible to fifth graders, such as, "the teaching method used in this course inspires my curiosity" [18]. A formal

assessment of the groups of students and statistical analysis of the students' responses in the questionnaire were performed to determine their positive or negative motivation.

(2) Assessment of Learning Effectiveness

The assessment of learning outcomes in this study consists of formative and summative measures. The formative assessment scripts are selected from the CD-ROMs of the publisher's question bank, and the average of the 50 multiple-choice questions are then examined and reviewed by several natural science teachers with more than 20 years of teaching experience. The summative assessment is in line with the school's examination schedule and process, and the school teachers were arranged by the school to carefully design the confidential examinations. Thereafter, other full-time science and technology teachers reviewed the questions and agreed to the assessment questions for the validity and credibility of the school examination. The assessments were kept confidential until the school administrative units had unified the printing prior to the Assessment of Learning Effectiveness.

(3) Interview and Statistical Methods

To understand the problems, feelings, and emotional changes of the research participants during the experimental teaching, the researchers compiled the outline of the tabletop game interview. The purpose of the interview process was to collect students' opinions and suggestions on the integration of tabletop games into the curriculum, in addition to other relevant information. The semi-structured interview was conducted using the following interviewing questions:

1. Do you like learning about natural science through tabletop games?
2. Do you think the integration of tabletop games has improved your learning?
3. Did you encounter any difficulties while playing tabletop games?
4. If given the opportunity, would you still want to learn about nature and life technology through board games?
5. Other things you want to say to the teacher.

In addition, content analysis was employed to determine the presence and meanings of certain words, themes, or concepts within interview [27].

This study collected data from the assessment of learning outcomes and interviews. The data from the interviews and the audio recordings were used to analyze the qualitative data of the interviews. The data obtained from the assessment results were processed using the SPSS 20 for Windows.

**3. Result**

In the following sections, we discuss the statistical analyses and qualitative data related to learning motivation, learning effectiveness, and the interview content.

*3.1. Independent Sample Testing of the Experimental and Control Groups*

One of the purposes of this study is to investigate the influence of tabletop games—integrated into the nature and life sections of the science and technology course—on students' learning motivation.

(1) Overall Motivation

To understand the changes in the performance of the experimental and control groups on the LMS after the integration of tabletop games into the curriculum, this paper first conducted an independent sample $t$-test based on the overall pre- and post-test results of learning motivation.

Table 4 shows the results of the independent sample $t$-tests of the experimental and control groups in the ARCS pre-test of the overall learning motivation. No significant difference was identified between the two groups in the pre-test performance. In other words, students' learning motivation toward the Sun and the world of plants was similar before the integration of tabletop games into teaching.

**Table 4.** Independent sample *t*-test of ARCS whole learning motivation pre-test.

| Motivation | Group (Number) | Mean | Standard Deviation | *t* Value |
|---|---|---|---|---|
| Learning motivation (Pre-test) | Control group (*n* = 52) | 116.04 | 21.57 | −0.311 |
| | Experimental group (*n* = 48) | 117.46 | 24.13 | |

Next, the experimental and control groups were tested in the ARCS overall learning motivation post-test independent sample *t*-validation; the results are presented in Table 5 below.

**Table 5.** The Independent Sample *t*-Test after the Overall Learning Motivation of ARCS.

| Motivation | Group (Number) | Mean | Standard Deviation | *t* Value |
|---|---|---|---|---|
| Learning motivation (Post-test) | Control group (*n* = 52) | 121.54 | 18.59 | −4.447 *** |
| | experimental group (*n* = 48) | 135.69 | 12.92 | |

Note: *** $p < 0.001$.

The statistical results in Table 5 show that in the ARCS overall learning motivation post-test independent sample *t*-test, the control group's mean is 121.54 and standard deviation is 18.59, and the experimental group's mean is 135.69 and standard deviation is 12.92. Accordingly, in the post-test part, the control group's mean is smaller than that of the experimental group. In addition, the statistical *t*-value is −4.447. The overall learning motivation of the experimental group was greater than that of the control group, which means that the incorporation of tabletop games had a positive effect on students' learning motivation. Furthermore, the interview data showed that the participants consider "The way we learn is more likely to attract our attention" (S5304).

(2) Overall Learning Effectiveness

To investigate the effectiveness of the integration of tabletop games into natural science teaching on the learning effectiveness of fifth-grade students, this section analyzes whether there are significant changes in the learning effectiveness of the experimental and control groups in the area of natural science after the integration of tabletop games into the curriculum.

The results in Table 6 reveal that in terms of the overall learning effectiveness, the mean and standard deviation of the control group are 251.98 and 25.83, respectively; the mean and standard deviation of the experimental group are 263.75 and 24.14, respectively, with a *t*-value of −2.349. Therefore, there is a significant difference in the learning effect between the two groups. Furthermore, in order to further understand the changes in the formative and summative assessments, this paper analyzed the learning outcomes of these three assessments individually.

**Table 6.** Overall learning outcomes of independent sample *t*-test.

| Evaluation | Group (Number) | Mean | Standard Deviation | *t* Value |
|---|---|---|---|---|
| Formative and summative assessment | Control group (*n* = 52) | 251.98 | 25.83 | −2.349 * |
| | Experimental group (*n* = 48) | 263.75 | 24.14 | |

Note: * $p < 0.05$.

The results in Table 7 clarify that among the three assessments of learning effectiveness, the first formative assessment of the experimental group had a significant difference, that is, Heart Attack and Back to Earth had significant positive impact on students' learning effectiveness. On the other two occasions, the experimental group's mean was slightly higher than that of the control group, but calculation using the statistical software clarified that there was no significant difference between the two groups in the second learning effectiveness. Therefore, the integration of the tabletop games Big Mouth and Operation

to Rescue Taiwan's Native Species into the curriculum did not significantly help students' learning effectiveness. Table 7 shows that the average score of the experimental group was higher than that of the control group in all three assessments, and the standard deviation was 2.24 less than that of the control group in the summative assessment, which clarifies that the experimental group's scores were higher and more concentrated after the integration of tabletop games.

**Table 7.** Individual learning effectiveness of independent sample *t*-test.

| Evaluation | Group (Number) | Mean | Standard Deviation | *t* Value |
|---|---|---|---|---|
| Formative assessment (The Sun) | Control group (*n* = 52) | 75.82 | 10.58 | −3.003 ** |
| | Experimental group (*n* = 48) | 82.71 | 12.35 | |
| Formative assessment (The World of Plants) | Control group (*n* = 52) | 88.41 | 9.47 | −1.551 |
| | Experimental group (*n* = 48) | 91.15 | 8.01 | |
| Summative assessment | Control group (*n* = 52) | 87.75 | 10.76 | −1.100 |
| | Experimental group (*n* = 48) | 89.90 | 8.52 | |

Note: ** *p* < 0.01.

### 3.2. Experimental Group Dependent Sample t-Test

Using Dr John Keller's ARCS model as the theoretical basis, the researchers adopted the ARCS to created four components: attracting attention, being relevant, building confidence, and creating satisfaction. The mean, standard deviation, and sample *t*-test were used to analyze the differences between the pre-test and post-test results of the four elements of the ARCS LMS that the students incorporated into the curriculum. The statistical analyses and explanations of the compiled data are shown in Table 8 below.

**Table 8.** Paired sample *t*-test of the pre-test and post-test dependencies of the experimental group on learning motivations.

| Element | Pre-Test | | Post-Test | | *t* Value |
|---|---|---|---|---|---|
| | Mean | Standard Deviation | Mean | Standard Deviation | |
| Attention | 30.60 | 6.12 | 35.88 | 2.82 | −6.76 *** |
| Relevance | 30.15 | 6.33 | 30.85 | 5.50 | −0.546 |
| Confidence | 29.54 | 7.06 | 29.58 | 6.91 | −0.027 |
| Satisfied | 30.63 | 7.41 | 31.63 | 5.28 | −0.703 |

Note: *** *p* < 0.001.

Table 8 reveals that the mean pre-test score of "Attract Attention" was 30.60 with a standard deviation of 6.12, and the mean post-test score is 35.88 with a standard deviation of 2.82 and a *t*-value of −6.76. Therefore, there is a significant difference between the pre-test and post-test, and the mean post-test score is larger than the mean pre-test score. The participants who were interviewed had said, "Concentration can last longer, and the functioning of the brain is much better".

This confirms that the inclusion of tabletop games can positively promote and support children's learning motivation in terms of "Attract Attention". In addition, the results show that the integration of tabletop games had a significant effect on the second and third assessments of learning effectiveness, however, Table 7 shows that the tabletop games did not offer much help to the students in terms of learning effectiveness in terms of summative assessment.

## 4. Discussion

There are two main research questions in this study, and the following is the discussion of these research questions. We also answer the research questions with the evidence of the qualitative analysis. Qualitative analysis was conducted for each tabletop game via

an interview after each of the four tabletop games, in which about 5 to 6 students were interviewed. The researchers added a letter "S" in front of the student's seat number to mark the responses of different students. In this section, we answered the research questions as follows. How do tabletop games integrate into the teaching of natural science, specifically the Sun and plants, in elementary schools? How do the integration of tabletop games into the teaching of natural science influence students' learning motivation and learning effectiveness?

*4.1. Research Question 1: How Do Tabletop Games Integrate into the Teaching Nature Science of the Sun and Plants in Elementary Schools?*

The four tabletop games designed by the instructor are as mentioned in above Table 3. "Heart Attack" is a children's game, "Back to Earth" is a party game, "Big Mouth" is a family game, and "Operation to Rescue Taiwan's Native Species" is a theme game. All of above games are easily understood by elementary school students. According to previous literature [28], the rules and game contents should not be designed to be too complicated or difficult. In addition, this paper highlights the importance of the relevance of the four tabletop games in the cognitive level of learning. We found that "Heart Attack" had low relevance in the cognitive level of learning, while "Back to Earth" was of high relevance in the cognitive level of learning in terms of natural science. In addition, "Big Mouth" and "Operation to Rescue Taiwan's Native Species" were of medium relevance in terms of knowledge of natural science. The result is echoed by a prior study [29].

Following a compilation of teacher's notes and student's feedback, it was determined that the tabletop games motivated the students to learn, especially "Heart Attack". The rules for "Heart Attack" were easy to understand and the game was easy to play, thus playing the game was an appropriate teaching strategy to motivate students or to focus their attention during the course. Previous research has documented that the integration of natural science into tabletop games can enhance students' motivation [30]. Therefore, elementary teachers could bring in easy game to motivate elementary school students in natural science education.

Secondly, the effectiveness of tabletop games on cognitive learning depends on the type of tabletop game. For example, the "Back to Earth" topic is designed to have a review effect, so it can reach a deeper level of cognitive learning by encouraging students to apply and analyze what they have learned, and thus enhance students' confidence and satisfaction, but this requires more time for children to think and play to achieve these results. Similarly, the card questions in "Big Mouth" can be designed as a complete sentences for true or false questions, or they can be designed with only the content of the questions from which the students must create their own questions. The latter encourages the students to reach the cognitive level of understanding so that they can formulate appropriate questions.

Finally, the cards in "Operation to Rescue Taiwan's Native Species" are already quite diverse, and the contents of the cards clearly indicate the propagation methods and native places of the plants. The rules were a bit more complicated in this game, so there is no need to make too many changes. Before or after playing this kind of tabletop game, we suggest that the teacher add some knowledge of native species or exotic plants to increase the knowledge and impression of students.

*4.2. Research Question 2: How Do the Integration of Tabletop Games into the Teaching of Natural Science Influence Students' Learning Motivation and Learning Effectiveness?*

The results showed no significant difference in the pre-test of learning motivation between the experimental and control groups, indicating that the two groups were not significantly different before the experimental design. However, there was a significant difference in the post-motivation test results. The average performance of the experimental group was higher than that of the control group, indicating that the students' learning motivation improved significantly after the integration of tabletop games into the curriculum.

In addition, there were no difference between the result of experimental and control groups on test 1 or test 2 in Table 7. However, the mean scores were higher in the exper-

imental group than that of the control group, and the standard deviation values within the groups showed that the experimental group was more concentrated than that of the control group, indicating that the integration of tabletop games is beneficial for students' focus. The interviews with participants clarify that tabletop games helped them concentrate better, improved their hand-eye coordination and mental agility, and improved their understanding of natural science. Lesson observations also showed that students were looking forward to the lessons, were willing to actively participate, and were willing to absorb more different ways of teaching natural science.

After the interviews, the researchers analyzed the four types of tabletop games in the ARCS motivation model, and the qualitative assessment of the interviews to understand the differences amongst students in different games, and whether the table games had different effects on the motivational elements of ARCS to help us better understand the connotations that cannot be explored in depth by quantitative statistics.

According to the content analysis of interview data, we found that all four games were highly correlated with motivation, especially in terms of fun and student interaction. In short, all four table games showed a high correlation amongst ARCS, which is consistent with the results of the above quantitative analysis.

During the four interviews with the children, the researchers found that the students were excited about learning the natural sciences curriculum through tabletop games in class. Student S5304 stated that "learning by playing attracts our attention more". In particular, some students said that the "Heart Attack" game helped them to be more responsive (Student S5412) and improved their concentration (Student S5410).

Some of students had more opportunities to interact with their classmates, while others felt that they had a happy mood and unforgettable courses. Student S5410 mentioned that: "When we were playing "Heart Attack", two of the students made a mistake, so we discussed a compromise and agreed on a solution and continued to play." Furthermore, playing tabletop games brought a good opportunity to collaborate with classmates, as the following statement suggests: "While playing Rescue Taiwan's Native Species, my classmates and I colluded to play strong" (S5413).

From the interviews, we can see that most of the students were very excited about the tabletop games and their whole hearts were tied to the games during the process. In their minds, tabletop games are not only tabletop games, but they are also a tool for communication, an object to enhance learning, a tool to promote students' relationship, and a better facilitator to understand each other's strengths and personalities. Therefore, the integration of tabletop games into the curriculum adds a lot of points to the motivation and emotion of learning, which is also in line with prior literature showing that students have given great reactions to the use of tabletop games in teaching [27].

However, the results of the experiment showed that the integration of tabletop games did not have a significant effect on the second and third formative assessments of learning effectiveness. The students had many unexpected gains and feelings during the interviews. Student S5405 mentioned that, "I learned about many planets, stars, and star clusters, etc. I felt a sense of accomplishment and the questions were easy to answer". Student S5321 told us that, "Playing "Back to Earth" can achieve the effect of review, broaden my knowledge, and strengthen my memory". Similarly, S5402 stated that, "I can guess the strength of the opponent and know what he does not understand in the game".

In summary, the above qualitative analysis corresponded to the statistical analysis of the study results. The result are also is line with prior research that regards educational tabletop games as a motivating force [3,25,31]. Tabletop games have played an important role in attracting learning attention, but they did not significantly affected students' learning outcomes.

## 5. Limitations of the Study

The limitations of this paper was related to the sampling strategies. As the sample size of four fifth-grade classes is not large, the convenience sampling strategy is not appropriate to extend the findings to other students from other districts or different level of education.

The school's administrative support and environment factors, such as classroom climate and cultures, were not taken into consideration, nor was the effect of the teaching methods.

## 6. Conclusions

The four tabletop games used in this paper tend to increase motivation, but they were less able to establish students' confidence, personal relevance, and satisfaction for the materials they were studying. For example, the tabletop game "Big Mouth" answers riddles that are closely related to the structure or function of plants in their surrounding environments, allowing students to be more integrated into the curriculum. The research design of this study needed to consider factors, such as time and curriculum content, in the design of the games, as the learning time for each tabletop game was only 20 min. The learning and the rules needed to be simple and easy to understand; thus the design of these tabletop games was biased towards the knowledge dimension in Bloom's cognitive level of learning, while the understanding dimension was only a small part, and other higher levels of learning that require more time in terms of the dimensions of application, analysis, and synthesis, were less involved. This research enables us to illustrate how tabletop game improve ARCS by putting forward a motivation-driven teaching approach.

The authors of this paper suggest that teachers take the phase of learning motivation into consideration while developing more types of tabletop games that can be integrated into natural science courses. Future studies are expected to have more extensive analysis and include students from different districts, different learning ages, different learning capabilities, and different learning programs. In addition, more studies are expected to illustrate how students with different motivations perceive various aspects of tabletop games integrated into their courses.

**Author Contributions:** Conceptualization, C.-Y.L., H.-H.L. and C.-C.L.; Methodology, K.-C.T.; Formal analysis, H.-H.L. and C.-C.L.; Investigation, C.-C.S.; Resources, C.-Y.L., H.-H.L. and C.-C.S.; Data curation, K.-C.T., H.-C.H. and L.-J.T.; Writing—original draft, H.-H.L.; Writing—review & editing, C.-C.L.; Visualization, C.-C.L.; Supervision, C.-C.L.; Project administration, C.-C.L. All authors have read and agreed to the published version of the manuscript.

**Funding:** The authors received no financial support for the research, authorship, and/or publication of this article.

**Data Availability Statement:** The data sets used and analyzed during the current study are available from the corresponding author on reasonable request.

**Conflicts of Interest:** The authors declare no potential conflict of interest with respect to the research, authorship, and/or publication of this article.

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
