# Peer review of "A Study on the Influence of Recreational Activities Intervening in Natural Science Courses on Learning Motivation and Learning Outcomes—The Case of Tabletop Games"

_sustainability, doi:10.3390/su15032509_

Round 1

Reviewer 1 Report

Good work, interesting topic, good prospects.

A couple of questions:

1. I recommend conducting a bibliometric analysis to determine the topic's relevance.

2. To my mind, the authors mean more learning incentives than motivation. Please provide a clear description of the incentives and motivation.

3. What are the practical recommendations resulting from research at this stage? The results are obtained and analyzed, but the model for implementing the results is not clear.

4. Can the result of the research be implemented for other levels of education? Please clarify.

All the best

Author Response

First reviewer

  1. Comment

Good work, interesting topic, good prospects. I recommend conducting a bibliometric analysis to determine the topic's relevance.

Response

  • Thanks a lot!
  • We revised tile and added new reference

  1. Comment

To my mind, the authors mean more learning incentives than motivation. Please provide a clear description of the incentives and motivation.

Response

  • We revised a paragraph (Line 74-84) make a clear definition
  • “Motivation is considered the psychological driving force behind human behavior (Locke, 1991; Katzell & Thompson, 1991) and the internal process of guiding an individual toward a goal (Roure & Lentillon-Kaestner, 2021). Motivation not only drives an individual to engage in a particular activity; but also provides an incentive for an individual to face the challenge (Hulleman et al., 2008; Roure & Lentillon-Kaestner, 2021). Since learning motivation covers a wide range theory, such as the attribution theory, and the self-efficacy theory (Lin, McKeachie & Kim, 2003), learning motivation is defined as the intrinsic motivation that causes and maintains students to engage in coursework learning activities voluntarily (Lucas, Pulido, Miraflores, Ignacio, Tacay & Lao, 2010; Liu, & Lipowski, 2021). "

  1. Comment

What are the practical recommendations resulting from research at this stage? The results are obtained and analyzed, but the model for implementing the results is not clear.

Response

  • We revised section 6 conclusion [Line482-502]
  • “The four tabletop games used in this paper tend to be more motivated, but are less important in establishing students’ confidence, personal relevance, and satisfaction. For example, the tabletop game "Big Mouth" answers riddles that are closely related to the structure or function of plants surrounding environments, allowing students to be more integrated into the curriculum. This is because the learning time of each tabletop game only takes up 20 minutes, and the research design of this study is based on various factors such as time, curriculum content and rules need to be simple and easy to understand, so the design of tabletop game is biased towards the "knowledge" dimension in Bloom's cognitive level of learning, while the "understanding" dimension only takes up a small part, and other higher levels of learning that require more time in terms of the dimensions of application, analysis, and synthesis. To sum up, this research enables us to illustrate how tabletop game improve ARCS by putting forward a motivation-driven teaching approach. “
  • We also demonstrate the teaching planning and tabletop game design in Table 1 and Table 2

4.Comment

Can the result of the research be implemented for other levels of education? Please clarify

Response

  • We added a new section 5. limitation of the study
  • “5. Limitations of the Study
  • The limitations of this paper was related to the sampling strategies. As the sample size of four fifth-grade classes is not large, the convenience sampling strategy is not appropriate to extend the findings to other students from other districts, different level of education. The school’s administrative support and environment factors, such as classroom climate, cultures, were not taken into consideration, nor was the effect of the lecture teaching method."

Reviewer 2 Report

This article deals with an extremely relevant topic of technology intervention in learning. 

Overall, the article is interesting and enlightening. There are few recommendations to the authors that should be considered.

1. Literature review needs to be strengthened. The authors just mention that the literature is "scanty". But the authors should cite them with results.

2. The scales need to be presented within the text of the article. What were the questionnaire items? 

3. Discussion of results is weak. It must be expanded and also validated against past studies. 

4. Implications should be more specific.

Author Response

Second reviewer

  1. Comment

This article deals with an extremely relevant topic of technology intervention in learning.  The article is interesting and enlightening. There are few recommendations to the authors that should be considered.

Response

Thank you for your kind help

  1. Comment

Literature review needs to be strengthened. The authors just mention that the literature is "scanty". But the authors should cite them with results.

Response

  • We re-restructured sections and new sections as follows (please see red color text)
  • We added a new reference Bawa, A. (2022). The Quest for Motivation: Tabletop Role Playing Games in the Educational Arena. International Journal of Game-Based Learning (IJGBL), 12(1), 1-12.

  • Introduction
  • 1 Learning Motivation and Teaching Strategies
  • 2. The Effectiveness of Integrating Tabletop Games in Nature science
  • 3. Aims and Research Questions
  • Research Method
  • 1. Research Design
  • 2. Research framework
  • 3. Research participants
  • 4. Tabletop Game Teaching Design and Research Process
  • 5. Instrument
  • Result
  • 1. Basic Statistical Data Analysis
  • 2. Independent Sample Testing of the Experimental and Control Groups
  • 3. Experimental Group Dependent Sample T-test
  • Discussion
  • 1. Research question 1
  • 2. Research question 2
  • Limitations of the Study
  • Conclusions

   3.Comment

The scales need to be presented within the text of the article. What were the questionnaire items? 

Response

  • We revised a paragraph in section 5. Instrument and added a new reference
  • Chang, Y.-H., Song, A.-C., & Fang, R.-J. (2018). Integrating ARCS Model of Motivation and PBL in Flipped Classroom: a Case Study on a Programming Language. Eurasia Journal of Mathematics, Science and Technology Education, 14(12), em1631. https://doi.org/10.29333/ejmste/97187

  1. Comment

Discussion of results is weak. It must be expanded and also validated against past studies. 

Response

  • We revised and added new Section 5. Discussion [page 11-page15, Line352-464]
  • “There are two main research questions in this study, and the following are the discussion of the research questions.--------------------------------------------------------------”----. “

5.Comment

Implications should be more specific.

Response

  • We revised section 6 conclusion [Line482-502]
  • “The four tabletop games used in this paper tend to be more motivated, but are less important in establishing students’ confidence, personal relevance, and satisfaction. For example, the tabletop game "Big Mouth" answers riddles that are closely related to the structure or function of plants surrounding environments, allowing students to be more integrated into the curriculum. This is because the learning time of each tabletop game only takes up 20 minutes, and the research design of this study is based on various factors such as time, curriculum content and rules need to be simple and easy to understand, so the design of tabletop game is biased towards the "knowledge" dimension in Bloom's cognitive level of learning, while the "understanding" dimension only takes up a small part, and other higher levels of learning that require more time in terms of the dimensions of application, analysis, and synthesis. To sum up, this research enables us to illustrate how tabletop game improve ARCS by putting forward a motivation-driven teaching approach. “
  • We also demonstrate the teaching planing and tabletop game design in Table 1 and Table 2
  • We added a new section 5. limitation of the study
  • “ Limitations of the Study The limitations of this paper was related to the sampling strategies. As the sample size of four fifth-grade classes is not large, the convenience sampling strategy is not appropriate to extend the findings to other students from other districts, different level of education. The school’s administrative support and environmental factors, such as classroom climate, cultures, were not taken into consideration, nor was the effect of the lecture teaching method.

Reviewer 3 Report

The abstract suggests that the paper fits into the special issue of Sustainability (Sustainability in Educational Gamification). Gamification in science education is an interesting topic. The novelty of the research is also lies in the fact that the authors have developed games specifically related to the curriculum. The research conducted by the authors promises to be comprehensive and thoughtful. However, the study needs major revision in terms of content, structure and English language and style. This is essential for further decision making.

Author Response

Third reviewer

1.Comment

The abstract suggests that the paper fits into the special issue of Sustainability (Sustainability in Educational Gamification). Gamification in science education is an interesting topic. The novelty  of  the  research  is  also  lies  in  the  fact  that  the  authors  have  developed  games specifically related to the curriculum. The research conducted by the authors promises to be comprehensive and thoughtful.

Response

  • Thanks a lot for your kind help

2.Comment

The study needs major revision in terms of content, structure and English language and style. This is essential for further decision making.

Response

  • Thanks a lot for your help. The paper has been proofread to remove typos and spelling errors.

3.Comment

However, The structure of the study needs to be revised. Two examples:-The research questions are included in the introduction. -There is a strange chapter title: ”Research tools”. This chapter includes a subsection on results and discussion. It is recommended that researchers follow the structure of scientific papers.e.g.,2https://www.mdpi.com/2079-3200/10/4/89

Response

  • We revised chapter title as 5 Instrument
  • Follow your suggestion https://www.mdpi.com/2079-3200/10/4/89 We re-restructured sections and new sections

  1. Introduction

1.1 Learning Motivation and Teaching Strategies

1.2. The Effectiveness of Integrating Tabletop Games in Nature science

1.3. Aims and Research Questions

  1. Research Method

2.1. Research Design

2.2. Research framework

2.3. Research participants

2.4. Tabletop Game Teaching Design and Research Process

2.5. Instrument

  1. Result

3.1. Participants’ background

3.2. Independent Sample Testing of the Experimental and Control Groups

3.3. Experimental Group Dependent Sample T-test

  1. Discussion
  2. 1. Research question 1
  3. 2. Research question 2
  4. Limitations of the Study
  5. Conclusions

4.Comment

The description of the ARCS model is very brief in the literature review section. However, the research is based on it.It is also unclear what criteria were used to assign a game to an element of the ARCS model. Although the authors mention that there have been numerous studies on the application of the ARCS model, they do not position their own research in this context

Response

  • We revised and added 1 Learning Motivation and Teaching Strategies

1.1 Learning Motivation and Teaching Strategies

Motivation is considered the psychological driving force behind human behavior (Locke, 1991; Katzell & Thompson, 1991) and the internal process of guiding an individual toward a goal (Roure & Lentillon-Kaestner, 2021). Motivation not only drives an individual to engage in a particular activity; but also provides an incentive for an individual to face the challenge (Hulleman et al., 2008; Roure & Lentillon-Kaestner, 2021). Since learning motivation covers a wide range theory, such as the attribution theory, and the self-efficacy theory (Lin, McKeachie & Kim, 2003), learning motivation is defined as the intrinsic motivation that causes and maintains students to engage in coursework learning activities voluntarily (Lucas, Pulido, Miraflores, Ignacio, Tacay & Lao, 2010; Liu, & Lipowski, 2021).

The ARCS Motivational Teaching Model was developed by Keller (1987) as an integrated motivation theory and instructional design to integrate the expectancy-value theory, the attribution theory, the achievement theory, etc. The learning motivation theory should also include effectiveness, efficiency, and engagement (Keller, 2008). The model focuses on the assimilation of an individual’s internal factors (e.g., personal values, abilities, and cognitive values) and factors external to the learning environment (e.g., instructional design). The ARCS model was well-developed and model is widely used (Keller, 1987; Li, & Keller, 2018).

The four components of the ARCS Model include Attention, Relevance, Confidence, and Satisfaction (Keller, 1987). The motivation model should first "arouse students' attention and interest in the course, then enable the students to discover that the course is personally relevant to them, so that students have confidence in themselves. Then students are capable of completing class task and attaining learners’ satisfaction (Keller, 2008). However, the actual teaching environment is extremely dynamic, and the individual differences of students are more important than the single teaching mode. Therefore, when using the ARCS motivation model, educators should assess students' needs in the light of the actual situation, particularly in the era of E-learning design (Keller & Suzuki, 2004).  The design of course curriculum should be designed according to students' needs so as to motivate them and maintain their learning progress. There is a wealth of literature on the application of the ARCS motivation model to teaching, and it has been applied to a variety of domains and stages. In this study, we mainly use the ARCS motivation model as a research model to explore the motivation patterns of school children.

5.Comment

The definition of board games is imprecise. It lacks the reference that makes it primarily defined as a card game. In fact, it cannot be defined mainly as a card game.

Response

  • We revised the line 48-55
  • We revised sub tile “the case of tabletop games” and united a term as “tabletop game”

6.Comment

The research questions are too general. In fact, the answer to the first question can be found in the research design. The question might be what measurable impact has been achieved by integrating these games into education. The study does not provide  systematic,  well-articulated answers to the research questions

Response

  • We added a new section 1.3 to cleaify the research questions
  • We revised and added new Section 5. Discussion [page 11-page15, Line352-464] by answering each question

1.3. Aims and Research Questions

Board games generally are used for recreational activities. In education, games enable students to have a better understanding of the relationship among content, process and             

context of a subject matter (Klabbers, 2003). Games in education are highly related to participants’ learning motivation and teachers’ teaching strategies. Accordingly, the aim of this paper is to investigate learning motivation with the effectiveness of integrating tabletop game. The research questions are as follows. How do tabletop games integrate into the units The Sun and the World of Plants in the course units of nature science? How do the influence of the integration of tabletop games into the teaching of nature science on students' learning motivation and learning effectiveness?

7.Comment

The process of pre-test compilation is described in detail by the authors. However, the number of teachers and experts who checked the instrument is not a good indicator. At least the reliability of the instrument should be reported.

Response

  • We revised Line 168-173 and report Cronbach alpha
  • We added a new reference Chang, Y.-H., Song, A.-C., & Fang, R.-J. (2018). Integrating ARCS Model of Motivation and PBL in Flipped Classroom: a Case Study on a Programming Language. Eurasia Journal of Mathematics, Science and Technology Education, 14(12), em1631. https://doi.org/10.29333/ejmste/97187

8.Comment

The methodology for recording and processing interviews is missing. The results of the interviews are presented here as just a few quotes from some interviews. The presentation of the quantitative results is very confusing, lacking a clear structure, making them difficult to evaluate.

Response

  • We revised the section [ (3) Interview and Statistical Methods, Line 253-263] and added interviewing questions
  • We revised and added new Section 5. Discussion [page 11-page15, Line352-464]
  • “There are two main research questions in this study, and the following are the discussion of the research questions-------------------------------------------------------------.“

9.Comment

The results reported are based on t-tests only. More in-depth analyses, possibly including background variables, are not available in the paper. The text regularly summarizes the statistical data included in the tables. This should be replaced by an interpretation of the results. The presentation of results and their interpretation (discussion) should be discussed in separate chapters.

Response

  • We revised and added new Section 5. Discussion [page 11-page15, Line352-464]
  • “There are two main research questions in this study, and the following are the discussion of the research questions-------------------------------------------------------------.“

10.Comment

In the first table, Remarks is not the correct name for the last column. The first figure suggests that the control group also benefited from the intervention.

Response

  • We deleted the remarks and revised a Table 1

  1. Comment

The fourth column of the second table (Teaching design) is redundant. I think, it contains no meaningful information. At the same time, the study does not reveal what the games were all about, or the information is not clear enough.

Response

  • We delete the fourth column and revised a Table 2

  1. Comment

The third table contains the background variables, but these data do not play any role in the analysis. However,  the  data  reported  in  this  table  is  not the  result  of  the  research,  it  is  not described in the relevant chapter. It contains only descriptive data. The authors refer to a table found in the previous chapter. This should be avoided.

Response

  • We changed the section title as 3.1. Participant’s background. The participants’ background is related to our research.

13.Comment

Overall, the study in its current form is extremely difficult to read. This is partly due to the linguistic quality (e.g. research questions themselves, 205-207, 243-245, 345-347), partly to the structure and partly to the content. The message of the  study is  not clear. Given the relevance of the topic and the potential for novelty, the study should be reconsidered after a major revision.

Response

  • Thanks a lot for your help. The paper has been proofread to remove typos and spelling errors.
  • Thank again for your kind comments and help.

Round 2

Reviewer 1 Report

Well done

Author Response

Thank you for your suggestion.
We trust that the manuscript has been improved with your assistance.
We have sent professionals to assist in English editing and submit files.
And share with you that we are enjoying Chinese New Year joy.

Reviewer 3 Report

The authors have provided adequate answers to most of the questions and the study has been mostly revised accordingly. Congratulations on the work so far, which shows thoroughness.

However, further improvements are needed. I have attached the document.

Author Response

Dear Reviewer

We would like to thank you for your great effort and patience to make this paper much better. We do our best to have a point-by-point response as follows

Best Regards

  • Point 1

In 1.1 Learning Motivation and Teaching Strategies:

There is a wealth of literature on the application of the ARCS motivation model to teaching, and it has been applied to a variety of domains and stages. – references still missing

Res:

1.We added following references

Chang, Y.-H., Song, A.-C., & Fang, R.-J. (2018). Integrating ARCS Model of Motivation and PBL in Flipped Classroom: a Case Study on a Programming Language. Eurasia Journal of Mathematics, Science and Technology Education, 14(12), em1631. https://doi.org/10.29333/ejmste/97187

Keller, J. M. (1987). Development and use of the ARCS model of instructional design. Journal of Instructional Development, 10(3),2-10. https://doi.org/10.1007/BF02905780

  1. Revised sentences

There is a wealth of literature on the application of the ARCS motivation model to teaching, and it has been applied to a variety of domains and stages (Chang, Song, & Fang, 2018; Keller, 1987 ).

  • Point 2

66: „The researcher intends to use tabletop games” – rather researchers intend? – if there are more researchers in the team

220: again: if there were only one researcher why do you use sometimes ’we’? If there were more researchers it would be better to change the singular to plural

Res:

  1. Thanks a lot for your help
  2. We replaced all “the researcher” as “researchers” or “ this paper” as line 65, 176, 194; 197;220;232;254;284;321;341; 361;387;432;441

  • Point 3

68-70: „Additionally, the timely incorporation of different tabletop game design teaching strategies to increase students’ learning motivation and effectiveness.”

design- as a verb: to make or draw plans for sth – My impression is that the incorporation cannot design anything – design and incorporate – both are verbs, are actions of smb. The sentence should be rephrased

Res:

  1. Thanks a lot for your correction
  2. We rephrased the sentence

“Additionally, the timely teaching strategies in terms of different tabletop games are expected to enhance students’ learning motivation and effectiveness.” (Line )67-68:

  • Point 4

134-135: “few studies have integrated tabletop games and units related to the sun or plants in the curriculum” – the sentence should be rephrased – integrated with? To integrate this and that into the curriculum is a bit strange.

Res:

  1. Thanks a lot for your correction
  2. We rephrased the sentence

“few studies have integrated tabletop game into a curriculum related to the sun or plants in an elementary school”

  • Point 5

135: However – ’despite this’ – but there is no contradiction – should be corrected

Res:

  1. Thanks a lot for your correction
  2. We replaced “However” with “Accordingly”

  • Point 6

Research questions:„How do tabletop games integrate into the units The Sun and the World of Plants in the course units of nature science?” – should be rephrased – What does it mean : into the units in the units?

„How do the influence of the integration of tabletop games into the teaching of nature science on students' learning motivation and learning effectiveness?” – should be rephrased – influence should be used as a verb.

Res:

  1. Thanks a lot for your correction
  2. We rephrased the research questions (Line 146-149)

How do tabletop games integrate into the teaching nature science of the Sun and Plants in elementary schools? How do the integration of tabletop games into the teaching of nature science influence students' learning motivation and learning effectiveness?

  • Point 7

168: at the begging? – at the beginning?

218: tabletop game – tabletop games (plural)

234-235: „The study used the Chinese version of the questionnaire and further revised by the researchers” – translated and revised by the researchers?

257: semi-structure interview – semi-structured

270: participant’s – participants’

311: he effect – the effect

390: „At the beginning of unit.” – should be deleted

394: lecturer – rather teacher (?)

396: the performance of tabletop games – performance? Find an appropriate word

417: has different – was different

432-433: „analyzed the four types of tabletop games in the 432 four motivational elements” – What does it mean

Res:

  1. Thanks a lot for your correction
  2. We changed all the typo and rephrased the sentence

  1. TYPO

168: at the begging? – at the beginning?

218: tabletop game – tabletop games (plural)

257: semi-structure interview – semi-structured

270: participant’s – participants’

311: he effect – the effect

390: „At the beginning of unit.” – should be deleted

417: has different – was different

  1. Rephrased the sentence

234-235: „The study used the Chinese version of the questionnaire and further revised --------”

394: elementary teachers

396: effectiveness of tabletop games

432-433: “the researchers analyzed the four types of tabletop games in the ARCS motivation model,----”

  • Point 8

Content:

I still dispute the clarity of Figure 1. The figure suggests that both groups played with the game. Especially since the explanation of the independent variable immediately after the figure says:„games … integrated into the curriculum for teaching” That is, as if to confirm that the games were  integrated into the teaching process of both groups. In the whole text (including the tables): use exeprimental group (not test group e.g. in Table 7) Please, check the following figure to see a no ambiguous representation of a similar process.

Res:

  1. Thank you for your clarification
  2. We have re-figured the figure 1 and make it clear
  3. We replaced test group with experimental group
  4. Re-named Figure 1. As Research framework

  • Point 9

273-274: „All the participants had played the tabletop game” – all? there is a contradiction with lines

210-213 „the control group did not play with the games

Res:

  1. Thanks a lot for your correction
  2. We rephrased the sentence as follows

-All the participants had” experience” of playing the tabletop game

-The control group did not play four types of tabletop games in the classroom.”

  • Point 10

3.1. chapter: should be placed in another chapter: this is not a result. These are descriptives statistics. These data describe the participants.

Res:

  1. Thanks a lot for your reminding
  2. We moved the section to section “2.3. Research participants”

  • Point 11

289-292:” No significant difference was identified between the two groups in the pre-test performance.

In other words, students' learning motivation was the same both before and after the integration of abletop games into teaching.” – refering to Table 4 representing the result of the pre-test. How do we know from the pre-test results what was the change before and after an intervention? The pre-test is conducted before the intervention.

Res:

  1. Thanks again
  2. We rephrased the sentence and make it clear

“In other words, students' learning motivation toward the Sun and the world of plants was the similar before the integration of tabletop games into teaching.”

  • Point 12

333: Table 7 shows (not Table 6)

Res:

  1. Thanks again
  2. We corrected the typo

  • Point 13

363-368: this is the description of the procedure – should be moved to the appropriate chapter

375-379: description of the games – this is not a discussion – should be moved to the appropriate chapter

381: there is no tables in the discussion (referred in the sentence)

396-406: this is not a discussion. This is the description of the games

A chapter about analysis of interviews should be included before Discussion.

The Discussion should discuss all results and not introduce new analysis (analysis of the interview).

Res:

  • We followed the 2 main research questions to discuss the result. For more clear description, we may repeat some information in the discussion section. Section 5. Discussion was revised [page 12-page14, Line489-370] as follows.  

  • “There are two main research questions in this study, and the following are the discussion of the research questions-------------------------------------------------------------.“

  • Point 14

422-423: „In addition, there was no significant difference between the second formative assess- 422 ment and summative assessment in Table 7” – there were no comparison of the results of the two tests. Rather: there were no diff. between the result of exp and cont groups nor on test 1 neither on test 2?

Res:

  1. Thanks again
  2. We rephrased the sentence and make it clear

“In other words, students' learning motivation toward the Sun and the world of plants was the similar before the integration of tabletop games into teaching.”

‘’In addition, there were no diffidence between the result of experimental and control groups nor on test 1 neither on test 2 in Table 7.”

  • Point 15

The method of analysing the interviews (qualitative analysis) is still missing from the study. Although, for example, in 437-438 we can read: „According to the analysis of interview data, we found that all four games were highly correlated with motivation

Res:

  • We revised the section [ (3) Interview and Statistical Methods, Line 262-272]  and added interviewing questions  and content analysis.
  • We added reference of content analysis as follows
  1. Stemler, S. (2000). An overview of content analysis. Practical assessment, research, and evaluation, 7(1), 17.
  • We rephrased the sentence : „According to the content analysis of interview data, we found that all four games were highly correlated with motivation
